# The impact of photovoice on mental health and stigma: A systematic review and meta-analysis

Adeniyi Adeboye[1☯], Ugochukwu Aghalu[1*], Whitney Onuorah[1‡],
Chizor Samuel-Nwokeji[1‡], Chimaobi Nwanguma[1], Adekunle Akerele[2], Judy Wasige[1☯]

1 Department of Nursing and Community Health, School of Health and Life Sciences, Glasgow Caledonian University, Glasgow, Scotland, United Kingdom, 2 Department of Epidemiology, Medical Statistics and Environmental Health, College of Medicine, University of Ibadan, Ibadan, Nigeria

☯ These authors contributed equally to this work.
‡ These authors also contributed equally to this work.
* UAGHAL300@caledonian.ac.uk

## Abstract

Photovoice is a transformative tool for improving health outcomes among people living with mental ill-health (PLWMI), fostering empowerment and social change. Only one meta-analysis has evaluated the effectiveness of photovoice for mental health outcomes. However, the study focused on various health conditions and only addressed two mental health outcomes. With an exclusive focus on mental health and stigma, this meta-analysis aimed to provide a comprehensive understanding of photovoice's impact on eight mental health and stigma-related outcomes in PLWMI and healthcare providers (HCPs). The study included seven randomised control trials (Sample size (n)=754), retrieved from electronic databases including Cochrane Trials, CINAHL Plus, ProQuest, Medline, PsychInfo, and Google Scholar, published between 2013 and 2023. Random-effects meta-analyses were conducted to calculate effect sizes (Cohen's d ($d$)). The study protocol was pre-registered on PROSPERO (CRD42023476867). Results showed significant improvements in mental health outcomes for PLWMI. Photovoice led to reductions in social withdrawal (n = 318; $d$, 95% CI = -0.20, -0.41 to 0.00) and depression (n = 267; $d$, 95% CI = -0.23, -0.45 to -0.02). Additionally, it increased self-efficacy (n = 267; $d$, 95% CI = 0.22, 0.00 to 0.44) and recovery (n = 285; $d$, 95% CI = 0.19, -0.02 to 0.41). However, there was a decrease in stigma resistance (n = 318; $d$, 95% CI = -0.13, -0.36 to 0.10). Among HCPs, there were significant improvements in both stigma and mental health-related outcomes, with reductions in anger (n = 330; $d$, 95% CI = -1.05, -2.79 to 0.69), social distance (n = 418; $d$, 95% CI = -1.41, -3.02 to 1.03), and the perception of danger (n = 115; $d$, 95% CI = -1.80, -5.04 to 1.44). The findings highlight the potential of photovoice as an effective intervention for improving mental health outcomes and reducing stigma in both PLWMI and HCPs. Consequently, photovoice can play a crucial role in

**Data availability statement:** All data utilised in this study has been fully provided in the manuscript and underlined supporting information files without exemptions.

**Funding:** This study was funded by a grant from See Me Scotland (URL: https://www.seemescotland.org/). There was no specific number associated with the grant. JW through ILFA Project Charity, was the specific recipient of the award that was designed as a community engagement project. See Me Scotland's understanding of the community engagement process provided an enabling environment that facilitated the integration of a research element to the community project. CN, WO, and CS received compensation from JW. The funders had no role in study design, data collection and analysis, decision to publish, or preparation of the manuscript.

**Competing interests:** The authors have declared that no competing interests exist.

enhancing the well-being of PLWMI and in transforming HCP attitudes within healthcare settings.

## Introduction

Photovoice lends a crucial perspective in healthcare as it enables individuals to creatively elaborate the nuances of their health issues and lived experiences [1]. Grounded in Goffman's symbolic interactionism, which posits that meanings are derived from people's interaction with symbols and other individuals, the use of narrative photography in research, especially through the photovoice method, has garnered widespread adoption in recent times [2]. Mainly, for its ability to uncover personal insights and deeper meanings behind visual representations of healthcare experiences [3,4].

Notwithstanding an increased awareness of mental health conditions, there is a deficit of knowledge on how people living with mental health issues (PLWMI) deal with the challenging stressors associated with their conditions amidst their daily lives. Jackson and Wasige [5] note that experiences of marginalisation, if rigorously assessed, often generate the knowledge required to counteract the marginalisation experienced. Subsequently, the paucity of knowledge about the experiences of PLWMI impedes the provision of relevant and adequate support [6]. According to extant studies, about 50–85% of the world population's mental health issues remain untreated or undiagnosed, resulting in a high global burden of the condition [7]. This was exacerbated by the COVID-19 pandemic which was reported to increase the severity of pre-existing mental health conditions and heighten the emergence of new cases in the general population, including among healthcare providers (HCP) [8–11]. The pandemic period also coincided with severe disruptions to mental health services, leaving gaps in care for those who needed it most while PLWMI are still unable to access relevant care and support post-pandemic [12,13]. Furthermore, the intentional avoidance of treatment due to the stigma from HCPs experienced by PLWMI is reported as one of the enabling factors of the increasing number of untreated cases [14]. Consequently, the explicit stigma demonstrated by HCPs towards PLWMI has been linked with the underutilization of mental health care services among this population [15,16]. Likewise, implicit stigma in HCPs has been related to reduced detection of mental illness cases due to inadequate assessment of potential cases brought on by prejudice and discrimination [16,17]. This stigma faced by PLWMI results in hopelessness, low self-esteem, and reduced empowerment, quality of life, as well as social support, which gravely interferes with recovery [18,19].

### Characteristics of photovoice

Photovoice interventions continue to improve health outcomes in diverse contexts thirty years after initial implementation [20]. Identified as participatory action research (PAR), the framework integrates collaborative research, education and action for social change facilitated through empowering marginalised communities to raise

awareness about their lived experiences [21]. PAR emerged from Freire's 'pedagogy of the oppressed' [22], which pre-supposes that a 'culture of silence' is enforced upon marginalised groups, which, predicated upon their prevailing circumstances keeps them submerged in a situation where demonstrating critical awareness and response becomes practically impossible. The implementation of photovoice is supported by Lewin's [23] conceptualisation of social problems as multi-disciplinary, and best addressed through 'action research', an iterative phased approach where the actions of one phase are informed by findings from previous phases [24].

The step-by-step collaborative approach begins with the enhancement of participants' photography skills to support effective documentation of their experiences and includes recruitment and an orientation session where the aims of the study, ethical considerations, and technical aspects of photography are discussed [21]. The value placed on participants' experiences as the building blocks for effective action, emphasises that such a process does not require external experts. The researcher acts as a facilitator of the photovoice process, guiding participants' discussions, encouraging deep reflection, and maintaining a non-judgmental space. Critically reflecting on the photographs taken through facilitated group discussions aims to enhance participants' awareness of the underlying causes of the issues in their community towards informing actions for community-level change. These discussions are crucial for co-constructing meaning and ensuring that the participants' voices are authentically represented [21]. Sharing these findings with policymakers and other interested stakeholders at exhibitions is often the final step that increases the traction behind social change [3,16,18].

Additionally, the participatory nature and focus on positive outcomes for communities reflects the principles of community-based participatory research (CBPR) which aims to centre on participants' expertise in conceptualising and developing suitable support to address their needs [20,25]. Participants are valued co-researchers throughout the research process, from developing research questions, collecting and analysing data, to disseminating the study's findings [26,27]. Maintaining fairness and equity in how power, control and trust are established and preserved across relationships in the research is emphasised by joint decision-making, democratising the research process [18]. Evidence shows that transferring power to participants is particularly therapeutic for individuals with mental health issues as it validates their experiences and provides a platform to voice their narratives [19]. This increases opportunities for the data collected to be relevant and useful to PLWMI and for complex problems to be addressed through harnessing the direct knowledge and expertise of the community's members, which positively influences the overall community health [3,28].

## Photovoice as an innovation

Over the years, increased efforts have been made to develop tools that would sufficiently support individuals with mental health conditions [25], while combating stigma, internalization of negative stereotypes, and enhancement of self-efficacy [29]. The historical under-investment in mental health services globally highlighted by the COVID-19 pandemic, prompted the urgent need for cost-effective interventions that are adaptable to wide-ranging contexts. Photovoice has been shown to have an inherent fluidity that enables its application in exploring various aspects of the lived experiences of PLWMI [20,30]. Appreciating participants' creativity and expertise increases opportunities for improving the practice of HCPs and has been shown to contribute to destigmatising mental illness. The decreasing cost of cameras in recent years and increasing accessibility to smartphones provide increased opportunities for capturing photovoice projects audio-visually. Combining this with the widespread use of social media makes Photovoice a cost-effective application that can be applied to diverse audiences, including in virtual settings.

However, various challenges have been identified with the photovoice approach in literature. Inconsistent adherence to the method, evidenced by published empirical articles lacking discussion on some aspects of the process has been cited [20]. For example, discussions about how photography training was implemented or how participants were involved in designing the research have been missed in some articles [31]. Challenges in ensuring that all participants are equally engaged and that power dynamics do not skew the interpretation of the visual data have also been reported [32]. Moreover, the analysis of photovoice data requires a nuanced approach that considers both the visual and textual elements,

often demanding interdisciplinary expertise [33]. Nonetheless, the limitations and benefits combine to provide a foundation for continued innovation and development of more equitable methods for mental health research and practice [34]. The insights derived from participants' photographs and subsequent discussions can inform the elicitation of more targeted and effective quantitative outcomes [35]. For example, integrating photovoice methodology into surveys or standardised assessment tools can help quantify changes in mental health outcomes, such as reductions in anxiety or improvements in self-esteem.

Thus, photovoice's developmental framework integrates creativity, flexibility, critical reflection and iteration to provide an intersectional approach to knowledge production that is adaptable to diverse settings. Importantly, it can be utilised in two major ways, functioning both as a powerful intervention for improving mental health and as a robust qualitative methodology for eliciting outcomes suitable for quantitative evaluation [36]. Its capacity to empower participants, raise consciousness, and foster community engagement, combined with its ability to generate rich qualitative data that can inform quantitative measures, makes photovoice a uniquely valuable tool in advancing mental health research and practice. Through this integrated approach, photovoice not only addresses immediate mental health needs but also contributes to long-term social change and policy development. Analysing the methodological underpinnings of photovoice, the data gathered can uncover significant variables and outcomes that might be overlooked in traditional mental health research [35]. For example, photovoice methodology has been used to identify community-specific issues, such as the need for better HIV/AIDS education amongst adolescents, which can then be quantified and addressed through targeted interventions [37].

The potential impact of photovoice on PLWMI extends beyond immediate mental health benefits to the achievement of recovery for the PLWMI. In recovery-oriented mental health care, the three dimensions of recovery include clinical, functional, and personal recovery [38]. Clinical recovery refers to the symptom-related aspects of mental illness, including the reduction of psychiatric symptoms and the improvement of overall mental health identified through clinical measures such as psychotherapy and relapse prevention [39]. Functional recovery emphasizes improvements in social activities, such as social relationships, employment, and self-sufficiency [40]. Personal recovery focuses on personal or individual experiences like empowerment, identity and hope, that reflect a person's ability to live a meaningful life despite the potential persistence of symptoms [41]. Subsequently, the adaptability and intersectionality of photovoice make it a powerful tool for advancing all aspects of recovery in PLWMI, offering a holistic approach to recovery that is both individualised and community-focused. This intersectional method allows for a multi-dimensional recovery process, where improvements in personal identity and social functioning enhance clinical outcomes, creating a synergistic recovery [38]. For example, as participants engage with photovoice and experience improvements in personal recovery through increased self-awareness and empowerment, they become more engaged in clinical treatment plans, improving clinical recovery outcomes. Simultaneously, as photovoice enhances social engagement, addressing functional recovery, individuals are better positioned to reintegrate into society, participate in social and vocational roles, and contribute to long-term policy change [21].

## Study rationale

Systematic reviews have explored the importance of photovoice in providing deep, qualitative insights into the lived experiences of individuals with mental health conditions [3,42]. Dong and Seo [42] conducted a review focusing on the experiences of people with severe mental illness utilising photovoice. The study highlighted the impact of photovoice on the individuals' recovery process, emphasising both positive experiences like improved self-esteem and challenges like experiencing frustration. Similarly, Han and Oliffe [3] explored participants' experiences of living with mental illness utilising photovoice in a scoping review. The review validated the significance of participant-produced images in gaining comprehensive insights into individual experiences of mental illness, with a particular emphasis on stigma and its impact on recovery. These studies emphasise the potential of photovoice to enhance recovery and address stigma in PLWMI. However, there is a noticeable gap in the literature regarding the quantitative evaluation of photovoice's impact on mental health outcomes and stigma reduction.

In an attempt to address the gap, a systematic review and meta-analysis investigating the impact of photovoice on a range of health conditions such as diabetes, hepatitis, physical health, social function, and mental health has been published [1]. However, the study analysed only two outcomes of mental health and did not consider stigma. Therefore, the current meta-analysis is the first study to elaborately focus on investigating the facets of mental health and stigma as they relate to PLWMI and their HCPs, with analysis of eight key outcomes: recovery, self-efficacy, stigma resistance, social withdrawal, depression, anger, social distancing, and perception of danger. Therefore, the novelty and relevance of the current study lies in the provision of a comprehensive overview of the effectiveness of photovoice in mental health and stigma from the perspective of both PLWMI and HCPs, which subsequently informs new evidence for practice.

In the context of this meta-analysis, mental health outcomes refer to outcomes that assess the internal psychic phenomena of patients, which cannot be externally examined or confirmed [43]. These measures typically assess either the frequency and severity of symptoms related to mental health illnesses (psychopathological rating scales) or evaluate the impact of mental health illnesses on an individual (measures of social functioning, or quality of life assessment) [43]. Furthermore, stigma-related outcomes refer to the various consequences or impacts that stigma has on PLWMI, and their ability to overcome such consequences [44]. These outcomes can affect multiple aspects of a person's life, including their mental health, social interactions, access to healthcare, and overall quality of life [44].

Consequently, this meta-analysis evaluates the impact of photovoice on mental ill health and stigma. It reviews appraised articles that report the effect of photovoice on the mental health outcomes of PLWMI such as depression, recovery, self-efficacy, and social withdrawal, as well as reducing negative behaviours exhibited by HCPs towards PLWMI, such as anger. Furthermore, the study also assesses the impact of photovoice on the enhancement of positive stigma-related outcomes in PLWMI, such as stigma resistance, in addition to the reduction of negative behaviours of HCPs that perpetuate stigma in PLWMI, such as the perception of danger and social distance.

The study looks to answer the question "What is the impact of photovoice on the mental health and stigma-related outcomes of PLWMI and HCPs"? To this effect, four hypotheses were formulated. Firstly, the authors hypothesised that photovoice has a significant effect on improving the mental health outcomes of PLWMI. Secondly, it was hypothesised that photovoice significantly reduces the negative mental health-related behaviours of HCP towards PLWMI. Thirdly, it was hypothesised that photovoice has a significant effect on improving the positive stigma-related outcomes of PLWMI.

Lastly, it was also hypothesised that photovoice significantly reduces the negative stigma-related behaviours of HCPs towards PLWMI. Therefore, this study analysed the effectiveness of photovoice on the mental health and stigma outcomes of PLWMI and the behaviours of HCPs towards PLWMI.

## Methods

This study employed a systematic review and meta-analysis to investigate the effectiveness of photovoice in improving mental health outcomes among PLWMI and the attitudes of their HCPs. It was pre-registered on PROSPERO with identification number - CRD42023476867. A systematic review provides a comprehensive summary of evidence regarding a specific research focus through a systematic approach to identify, appraise, and synthesize all relevant evidence [45,46]. Additionally, meta-analysis involves statistical analysis that combines and synthesizes the results of numerous studies to integrate their findings and produce a quantitative estimate [47,48]. This proffers increased accuracy in effect estimation, conflict resolution between studies of the same foci, and generalizability of research findings [49,50].

### Eligibility criteria

**Types of studies.** Primary quantitative experimental studies, exclusively randomized control trials or mixed methods studies incorporating a qualitative photovoice-based study as an intervention in a randomized controlled design, were

included [51,52]. The population of interest were PLWMI and their HCPs. No restrictions were implemented on the type of mental illness, age, or setting. Studies addressing mental health outcomes such as depression, recovery, self-efficacy, social withdrawal, stigma-resistance, anger, social distance, and perception of danger, and published in the English language between January 2013 and November 2023 were included.

Only studies with explicitly stated photovoice methodology, or participatory photography that entailed training participants on photography and facilitating focus-group discussions on photography and mental health were included. PLWMI would take photographic images that depicted their lived experiences of mental ill-health and interpret the photographs to the researcher to determine the meanings attributed. Participants were co-researchers and solely responsible for conceptualizing and allocating meaning to the pictures, as well as selecting photos and uncovering the stories behind them to the researchers [21,28,53].

Included studies with healthcare practitioners measured the effectiveness of photovoice in ameliorating negative attitudes towards PLWMI. Excerpts from photovoice participatory studies would be introduced to some healthcare practitioners as an intervention and compared with others where there was either no intervention or practice as usual on anti-stigma interventions.

## Search strategy and study selection

A scoping search was carried out on Cochrane Trials, CINAHL Plus, and Google Scholar to determine the availability of studies addressing the research question, inform the search development process, and aggregate keywords used in relevant articles [54]. Between October to November 2023, five academic databases– Cochrane Trials, CINAHL Plus, ProQuest, Medline, and Psych Info, were systematically perused for relevant papers (See S4 Appendix for the full list of all studies identified in the literature search). Google Scholar search engine was also incorporated to identify grey literature, and minimize publication bias [55,56], which promulgates comprehensiveness of the search process and evidence base, as well as minimizes omission of otherwise relevant studies which may not be present on generic databases [57,58]. To ensure further elimination of publication bias, specific journals renowned for publishing negative results, such as PLOS ONE and PLOS Global Public Health were also particularly targeted and reviewed [59]. Covidence automation software, alongside manual data management, was incorporated in this review to ensure completeness in the retrieval of all available data, transparency, and bias reduction in the review process. Two reviewers, CN and JW independently screened the titles and abstracts of identified studies against the eligibility criteria, and an agreement was reached on the articles for full-text screening. Additionally, the reference lists of relevant articles were manually scanned for relevant papers. Concomitantly, CSJ and WO utilized Covidence for citation screening, data extraction and quality appraisal, which facilitated the reliability and rigour of the review process [60]. The search process and study selection for inclusion were stringently guided by the Preferred Reporting Items for Systematic Reviews and Meta-Analyses (PRISMA) checklist (see S1 Appendix), for which a flow diagram was generated to aid transparency and reproducibility [61,62]. See S1 Table for the full search strategy conducted on all selected databases.

## Data extraction

Initial data extraction was carried out independently by two researchers, CSJ and WO on Covidence. A form (see S2 Table), was developed from manuals of the Centre for Reviews and Dissemination, Joanna Briggs Institute, and consultation with experts in the subject area [63,64]. This form was piloted on two included studies to ascertain suitability and adjusted as required [63]. Data was extracted in textual, numerical, and tabular formats, and a third reviewer, AA, was invited to resolve any conflicts. This was done to ensure quality control, completeness, and reliability of extracted data [65]. The initial data extraction focused on study characteristics (publication year, author, design, aim, sample size, survey instrument); participant characteristics (age, gender); intervention (photovoice and comparator); outcomes and effect sizes (mean, standard deviation, power calculation).

Furthermore, AA and AA extracted the relevant data specific for the meta-analysis into a Microsoft Excel Spreadsheet which was already prepared with appropriate and relevant columns detailing shortlisted studies that met the eligibility criteria. No conflicts were encountered at this stage of data extraction. Variables extracted from these studies included mean/SD of study outcomes as well as effect sizes where available. Extracted data were shared with the members of the research team for the second level of data verification before the analysis was conducted. The standard error of each effect size was estimated using the recommended formula as none was available in all eight eligible studies [66].

## Quality assessment

Quality appraisal was done using a validated tool for quantitative studies, the Effective Public Health Practice Project [EPHPP] [67] on Covidence. This assessed quality within six domains - selection bias; study design; confounders; blinding; data collection methods; withdrawals and dropouts. These result in an overall global rating, wherein studies with no weak ratings amongst the six domains are considered 'strong,' studies with one weak rating are considered 'moderate,' and studies with two or more weak ratings are considered 'weak.' Two reviewers, CSJ and WO, independently assessed quality, and discordances were resolved through discussion, or by a third reviewer, AA.

## Data analysis

A meta-analysis was conducted to pool studies with similar outcomes. This was done in two phases; one phase focused on studies on positive and negative mental health and stigma-related outcomes of PLWMI, and the other on studies on positive and negative mental health and stigma-related attitudes of HCPs towards PLWMI.

Quality control was done and maintained by members of the review team to ensure validity. Raw data (that is, mean and standard deviation of the two arms of each study) were used to calculate the effect size from all the studies included using the recommended formula [66]. This is a preferred approach to calculating effect size due to inconsistency in reporting the precalculated effect size across the extracted studies. This approach gives a concise estimate and direction of the effect size [68]. Based on the RCT study design of each eligible study with a focus on variable outcomes of interest, a within-sample estimation of photovoice effectiveness was calculated using the mean/standard deviation of each arm of each eligible study. Results reported included confidence intervals of the effect size of each outcome and the overall effect size of the study outcome (Cohen's d). Due to expected study design differentials in the eligible studies, a random effect model was used for the calculation of the overall effect size of photovoice on each outcome [69]. The use of the random effect model as opposed to a fixed effect model ensures that potential variabilities due to expected differentials in study design among eligible studies are minimized in the calculation of overall effect size. Z distribution of 95% confidence intervals (CIs) was employed to estimate overall effect size. This approach is said to offer a parsimonious and narrower CI to the study outcomes when one considers the available sample size of at least two studies per outcome [69]. Funnel plots were used to assess the publication bias of studies included in the analysis due to the sample size of available studies for each outcome.

The I-squared statistic ($I^2$) from forest plots was used to assess the heterogeneity of the precalculated effect size from various studies. Within-group heterogeneity was assessed using $I^2$ statistic with ≥ 50% threshold indicating substantial heterogeneity. The Forest plots were used to present the magnitude and direction of the overall effect size relative to the line of no effect. Estimation was done at a 5% significant level. SPSS version 22.0 was used to conduct all analyses.

## Data synthesis

Further data management, quality control and required analysis were carried out based on the recommended guidance [70]. Overall, four studies were considered to address the mental health-related outcomes in PLWMI. The outcomes considered included "Depression, Recovery, Self-Efficacy, and Social Withdrawal". Additionally, three studies were considered to address stigma-related outcomes in PLWMI, with "stigma resistance" being the only outcome evaluated. Furthermore, in addressing the mental health-related outcomes in HCPs, three studies that met eligibility criteria were available for

the analysis and one outcome, "anger" was evaluated for this analysis. Finally, addressing the stigma-related outcomes in HCPs necessitated the evaluation of three studies that met the criteria for two outcomes, "perception of danger" and social distance".

## Results

### Study selection

The database search identified a total of 331 potentially relevant studies which were imputed into Covidence for the removal of duplicates. At this stage, a total of 31 duplicates were removed, leaving a total of 300 studies for screening based on the titles and abstracts on Covidence. The titles and abstracts screening led to the removal of a further 283 articles, leaving a total of 17 articles for full-text reading. At this stage, two reviewers read the full text of the 17 articles to determine their adherence to the eligibility criteria in the current review. This led to an exclusion of 10 articles, leaving a total of 7 articles for quality assessment in this review. Reasons for exclusion included the identification of 2 studies as study protocols, 2 studies did not utilise photovoice-based interventions, and 6 studies were not randomised controlled trials. Fig 1 illustrates the study selection process for the current review. See Table 1 for the characteristics of the seven included studies. The list and references of all included studies are provided in S2 Appendix.

### Quality assessment results

As earlier indicated, the seven included studies were assessed for quality by utilising the EPHPP tool. The EPHPP tool on Covidence facilitated the extraction of relevant data for quality assessment.

Regarding selection bias, two studies were rated strong, while five studies were rated moderate. For the study design, all included studies were rated strong which is a shortcoming of the EPHPP tool as RCTs are automatically assigned a strong rating for this domain. Regarding the influence of confounders on the outcomes, two studies received a strong rating which indicated that there were no differences in baseline measures between intervention and control groups, while five studies were rated weak due to differences between groups at baseline. Only one study received a strong rating for blinding which indicated that the research question was concealed from participants and that outcome assessors were not aware of the intervention status of participants, while the remaining studies received a moderate rating for this category. Additionally, all studies utilised valid and reliable tools for data collection, with full details provided on how the tools were validated before data collection leading to a strong rating for all included studies in this domain. Regarding withdrawals and dropouts, two studies were rated strong because they reported the withdrawals fully, including reasons for the loss of participants and above 80% of study participants completed the studies. Additionally, 3 studies received a moderate rating due to recording a 75% study completion rate among study participants. Furthermore, 2 studies received a weak rating in this category. One of the studies recorded less than a 60% study completion rate, in contrast to the final study not including reports on withdrawals.

Overall, only one study was assessed as strong, indicating a high quality, five studies were assessed as moderate indicating moderate quality, while one study was assessed as weak indicating a weak quality. The overall results are illustrated in Table 2.

### Effect of photovoice on the mental health-related outcomes in PLWMI

**Depression.** Two papers provided data on the effect of photovoice on depression as an outcome of mental illness. The overall effect size was -0.23 (95% CI, -0.45 to -0.02), demonstrating a 23% reduction in depression following the use of photovoice. The forest plot of the effect size of photovoice on depression is presented in Fig 2.

**Recovery.** Three papers provided data on the effect of photovoice on recovery as an outcome of mental illness. The overall effect size was 0.19 (95% CI, -0.02 to 0.41), demonstrating a 19% chance of recovery from mental illness when photovoice is used. The forest plot of the effect size of photovoice on recovery is presented in Fig 3.

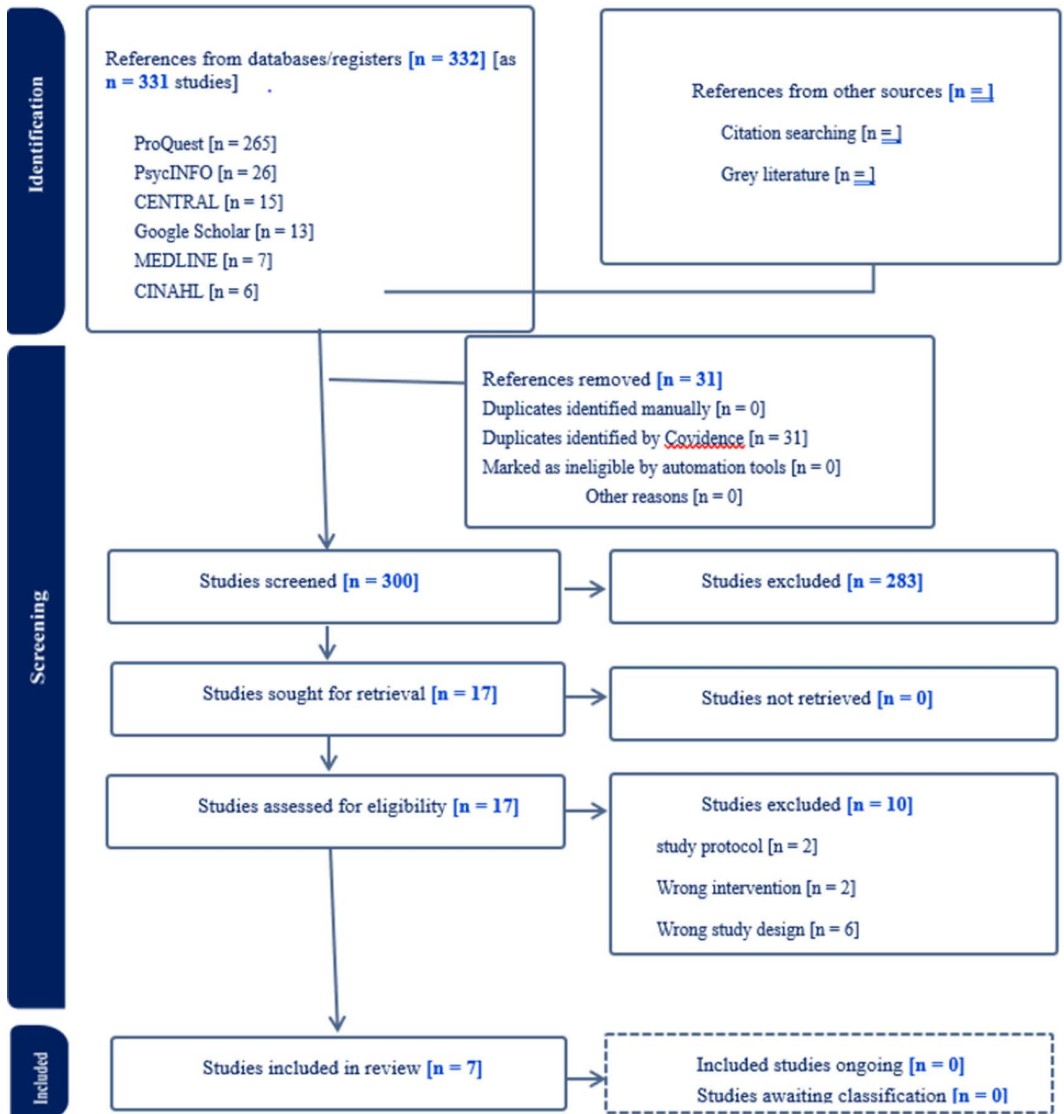

**Fig 1. PRISMA flow chart.**

**Self-efficacy.** Two papers provided data on the effect of photovoice on self-efficacy as an outcome of mental illness. The overall effect was 0.22 (95% CI, 0.00 to 0.44), demonstrating a 22% increase in self-efficacy after photovoice was used. The forest plot of the effect size of photovoice on self-efficacy is presented in Fig 4.

**Social withdrawal.** Three papers provided data on the effect of photovoice on social withdrawal as an outcome of mental illness. The overall effect size was -0.2 (95% CI, -0.41 to 0.00), demonstrating a 20% reduction in social withdrawal after photovoice was used. The forest plot of the effect size of photovoice on anger is presented in Fig 5.

### Effect of photovoice on the stigma-related outcomes in PLWMI

**Stigma resistance.** Three papers provided data on the effect of photovoice on stigma resistance as an outcome of mental illness. The overall effect size was -0.13 (95% CI, -0.36 to 0.10), demonstrating a 13% reduction in stigma

**Table 1. Characteristics of included studies.**

| Study/ Setting/ Country | Population | Design | Intervention | Comparison | Outcomes | Outcome measures | Gender/Age | Sample size | Results |
|---|---|---|---|---|---|---|---|---|---|
| 1.Zhang et al 2023 Hong Kong, China | PLWMI | Mixed Methods (quantitative analysis designed as an RCT). | Mindfulness-Based Family Psycho-education Intervention with a Photovoice component | Family Psycho-education (FPE) | Recovery | MHRM | Not Available. | 18 Young adults in recovery (YAIR) | Recovery level of YAIR increased after their caregivers participated in MBFPE |
| 2. Flanagan et al 2016 USA | HCPs | RCT | 1hr photovoice performance by PLWMI | No intervention | 1) anger 2) recovery 3) social distance | 1) Attribution Q. 2) RKI 3) SDS | 93% female / Mean age ± SD 49.36±10.5 | 27 HCPs | Significantly decreased negative stereotypes, attribution of dangerousness, fear, desire to segregate people from the community, and desire for avoidance. |
| 3. Russinova et al 2018 USA | PLWMI | RCT | Vocational Empowerment Photovoice (VEP) | No intervention | 1) social withdrawal 2) stigma resistance | ISMI | Intervention group – 66.7% female; Control – 55.6% female | 51 individuals with psychiatric disabilities | 1) Significantly higher rate of engagement in employment services 2) Increased work hope, self-efficacy and sense of vocational identity |
| 4. Russinova et al 2023 USA | PLWMI | RCT | Bridging Community Gaps Photovoice (BCGP)" program | Services as usual | 1) depression 2) recovery 3) self-efficacy 4) social withdrawal 5) stigma resistance | 1) BASIS24 2) PGRS 3) ICP-MH 4&5) ISMI | Intervention group – 60.6% female; Control – 61.5% female | 185 participants | 1) Increased community engagement 2) Reduced self-stigma Significant impact on self-efficacy |
| 5. Russinova et al 2014 USA | PLWMI | RCT | Peer-run photography-based intervention | No intervention | 1) Depression 2) Recovery 3) Self-efficacy 4) Social withdrawal 5) Stigma resistance | 1) Depression scale 2) PGRS 3) GPSES 4&5) ISMI | 68%>40yrs; 68% female | 82 PLWMI | 1) Significantly reduced self-stigma, greater, and perceived recovery and growth. 2) No differences between groups in depression, self-efficacy |
| 6. Kohrt et al 2021 Nepal | HCPs | Pilot Cluster RCT | RESHAPE training | Training as usual (MHGAP-IG) | 1) Perception of danger. 2) Social distance | 1) SDS 2) IAT | 85% male/mean age±SD 36.2 ± 8.8 years | 88 PCPs | Reduced stigma compared with standard training |
| 7. Tippin and Maranzan 2019 Canada | HCPs | RCT | Online antistigma photovoicebased video | Control video | 1) anger 2) social distance | 1) Attribution Q 2) SDS | Female: 73.9%; Male:26.1% Mean age - 21.44; SD (5.0) | 303 participants. | Efficacious in reducing mental illness stigma; reduced fear and anger toward PLWMI, decreased perceptions of dangerousness, and desired social distance. |

**KEY:** MBFPE - Mindfulness-based Family Psychoeducation; YAIR - Young adults in recovery; BASIS-24 – Behaviour and Symptom Identification Scale; GPSES – Generalized Perceived Self-Efficacy Scale; IAT – Implicit Attitude Test; ICP – MH – Inventory of Community Participation – Mental Health; ISMI – Internalized Stigma of Mental Illness; MHRM – Mental Health Recovery Measure; PGRS – Personal Growth and Recovery Scale; RKI – Recovery Knowledge Inventory; SDS – Social Distance Scale; MHGAP-IG- Mental Health Gap Action Programme-Intervention Guide; RESHAPE- Reducing Stigma Among Healthcare Providers to Improve Mental Health Services.

**Table 2. EPHPP tool quality assessment results.**

| | | Khort *et al* 2023 | Flanagan *et al* 2016 | Zhang *et al* 2023 | Russinova *et al* 2014 | Russinova *et al* 2018 | Russinova *et al* 2023 | Tippin and Maranzan 2019 |
|---|---|---|---|---|---|---|---|---|
| **A. SELECTION BIAS** | | | | | | | | |
| Q1 | Representative of the target population? | VL | SL | SL | SL | SL | VL | SL |
| Q2 | Percentage of selected that agreed to participate? | 100% | CT | CT | CT | CT | 80-100% | CT |
| Section Rating | | Strong | Mod | Mod | Mod | Mod | Strong | Mod |
| **B. STUDY DESIGN** | | | | | | | | |
| Q1 | Described as RCT? | Yes | Yes | Yes | Yes | Yes | Yes | Yes |
| Q2 | Randomization method described? | Yes | No | Yes | No | No | No | No |
| Q3 | Randomization method appropriate? | Yes | N/A | Yes | No | No | No | No |
| Section Rating | | Strong | Strong | Strong | Strong | Strong | Strong | Strong |
| **C. CONFOUNDERS** | | | | | | | | |
| Q1 | Important differences between groups before intervention? | No | CT | CT | No | Yes | CT | CT |
| Q2 | If Yes, percentage of confounders controlled? | N/A | N/A | N/A | N/A | CT | CT | CT |
| Section Rating | | Strong | Weak | Weak | Strong | Weak | Weak | Weak |
| **D. BLINDING** | | | | | | | | |
| Q1 | Outcome assessors aware of intervention status of participants? | Yes | Yes | CT | Yes | Yes | CT | No |
| Q2 | Study participants aware of the research question? | No | No | No | CT | CT | CT | No |
| Section Rating | | Mod | Mod | Mod | Mod | Mod | Mod | Strong |
| **E. DATA COLLECTION METHODS** | | | | | | | | |
| Q1 | Data collection tool valid? | Yes | Yes | Yes | Yes | Yes | Yes | Yes |
| Q2 | Data collection tool reliable? | Yes | Yes | Yes | Yes | Yes | Yes | Yes |
| Section Rating | | Strong | Strong | Strong | Strong | Strong | Strong | Strong |
| **F. WITHDRAWALS** | | | | | | | | |
| Q1 | Withdrawals and dropouts reported? | Yes | No | Yes | Yes | Yes | Yes | Yes |
| Q2 | Indicated the percentage of participants that completed the study? | 93% | N/A | N/A | 75% | 75% | 90% | <60% |
| Section Rating | | Strong | Weak | Mod | Mod | Mod | Strong | Weak |
| **Global rating for each study** | | **STRONG** | **WEAK** | **MOD** | **MOD** | **MOD** | **MOD** | **WEAK** |

**KEY:** VL – Very Likely; SL – Somewhat Likely; CT – Can't Tell; MOD – Moderate; N/A – Not Available; SL – Somewhat Likely; CT – Can't Tell; MOD – Moderate; N/A – Not Available. Adapted from: [67].

resistance after photovoice was used. The forest plot of the effect size of photovoice on stigma resistance is presented in Fig 6.

### Effect of photovoice on the mental health-related outcomes in HCPs

**Anger.** Two papers provided data on the effect of photovoice on anger as an outcome of mental illness. The overall effect size of photovoice on anger was -1.05 (95% CI, - 2.80 to 0.69). This demonstrates that photovoice reduced anger in study participants by 105% following its use. The forest plot of the effect size of photovoice on anger is presented in Fig 7.

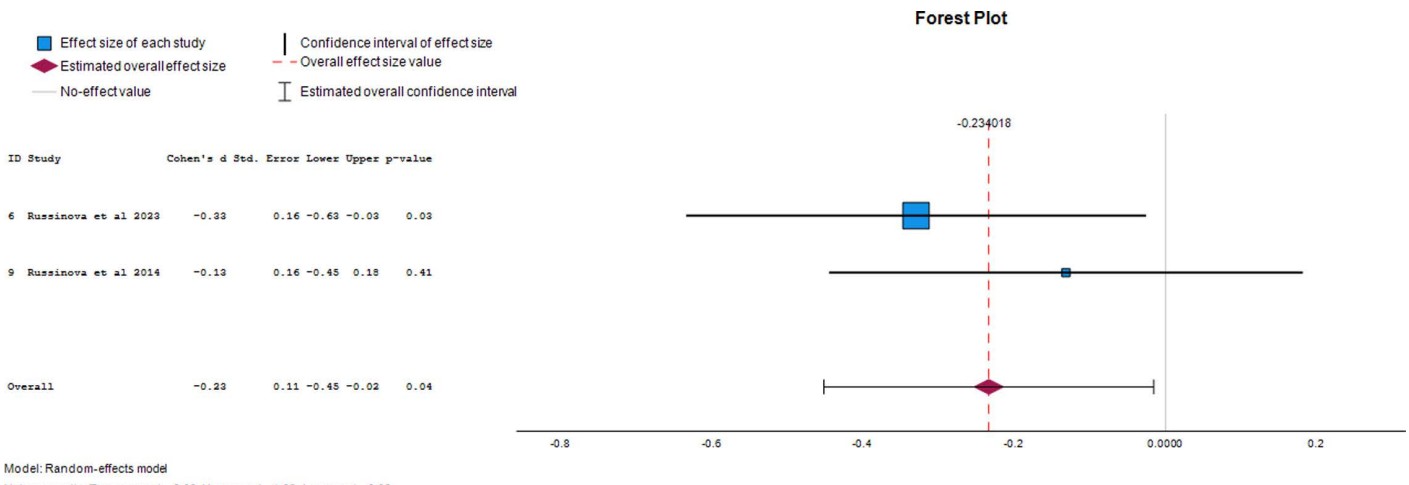

**Fig 2. Forest plot of effect size of photovoice on depression.**

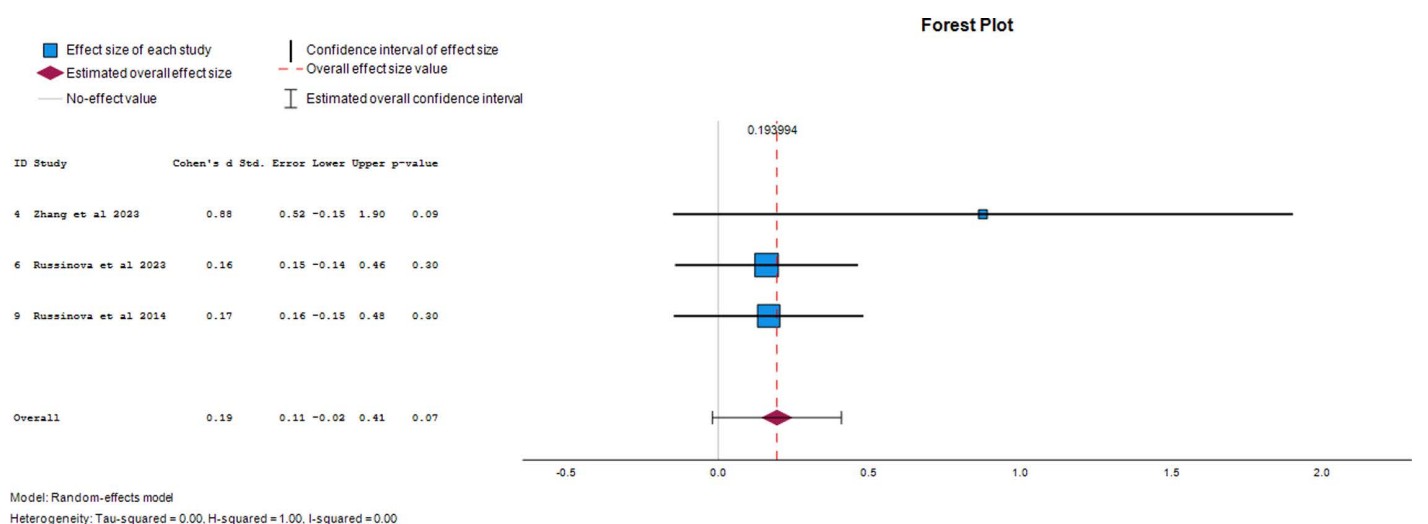

**Fig 3. Forest plot of effect size of photovoice on recovery.**

### Effect of photovoice on the stigma-related outcomes in HCPs

**Social distance.** Three papers were utilized to extract data on the photovoice-generated effect size for social distancing in HCPs. Overall, the effect size of photovoice on social distance in HCPs was -1.41 (95% CI, -3.02 to 1.03). This demonstrates that photovoice reduced social distance in study participants by 141% following its use. The forest plot of the effect size of photovoice on social distance is presented in Fig 8.

**Perception of danger.** Two papers were utilized to extract data on the photovoice-generated effect size for social distancing in HCPs. Photovoice had a -1.80 (95% CI, -5.04 to 1.44) effect size on the perception of danger in HCPs. This demonstrates that photovoice reduced the perception of danger in study participants by 180% following its use. The

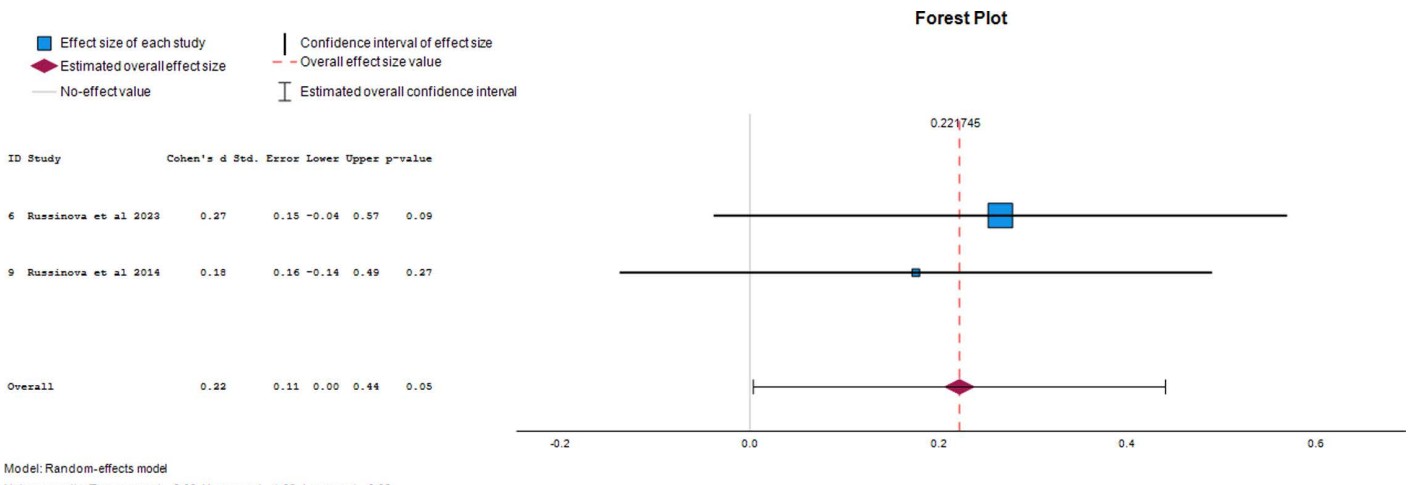

**Fig 4. Forest plot of effect size of photovoice on self-efficacy.**

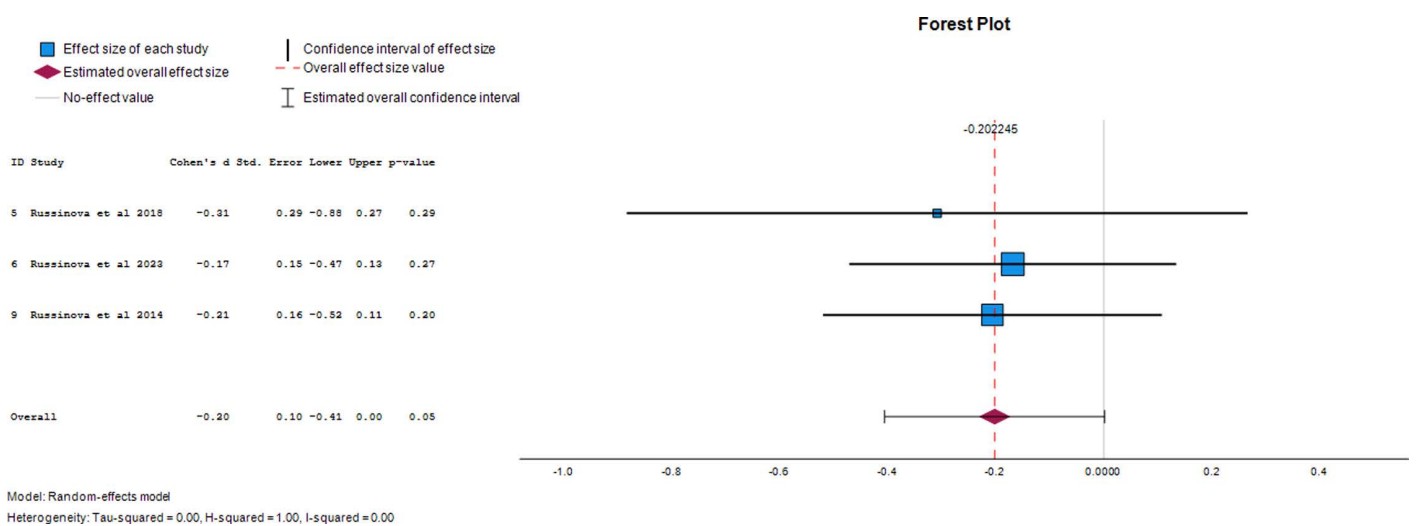

**Fig 5. Forest plot of effect size of photovoice on social withdrawal.**

forest plot of the effect size of photovoice on perception of danger is presented in Fig 9. **Table 3** details the effect sizes calculated from the available raw data for all outcomes.

## Publication bias

To assess potential publication bias, funnel plots were generated for each outcome analysed. The funnel plots for most outcomes displayed a relatively symmetrical distribution of study effects around the pooled estimate, suggesting a low risk of publication bias. However, asymmetry was observed in the funnel plots for mental health and stigma-related outcomes in HCPs. However, the variability in the sample sizes and the precision of studies included for these outcomes provide probable explanations for the observed asymmetry. For example, for anger outcome, there were 27 participants in the study conducted by Flanagan *et al.* compared to 303 in the study conducted by Tippin and Maranzan (See S3 Appendix).

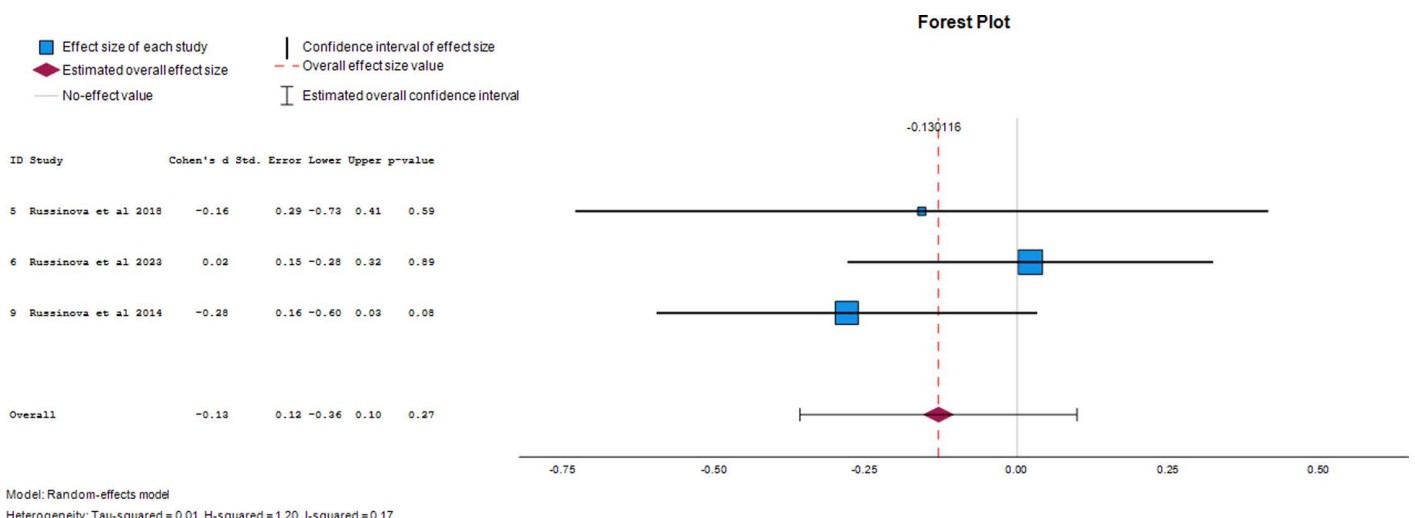

**Fig 6. Forest plot of effect size of photovoice on stigma resistance.**

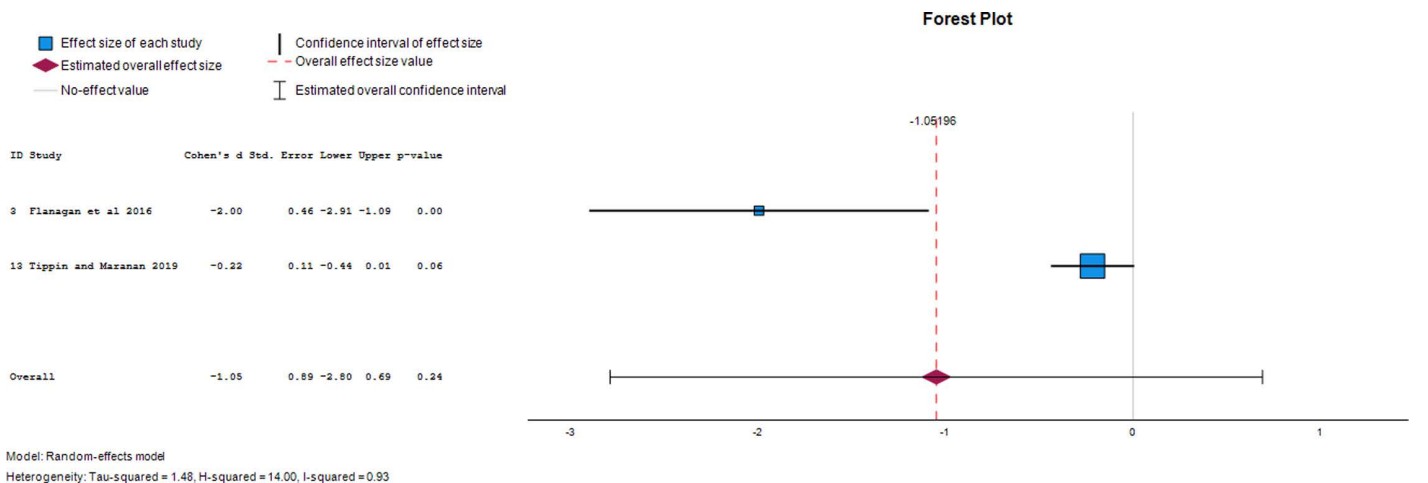

**Fig 7. Forest plot of effect size of photovoice on anger.**

## Discussion

The results of this meta-analysis suggest that photovoice is an effective tool for reducing negative mental health and stigma outcomes and improving positive outcomes in PLWMI. Conceptually, this is interlinked with synthesised outcome measures, including recovery, self-efficacy, and stigma resistance. However, debates among researchers on the notion of the true meaning of recovery for the individual living with mental illness continue [71,72], with three prominent perspectives: personal, clinical, and functional recovery.

Traditional mental health interventions such as clinical staging, a deficit-based strategy that aims to mitigate outcomes of mental illness for achieving clinical recovery have been effective in reducing clinical symptoms [73,74]. However, such interventions have been criticised for being reductionistic, and not focusing on the perspective of the PLWMI regarding the meaning of recovery [71]. In contrast, photovoice represents a personal recovery-oriented approach that emphasises

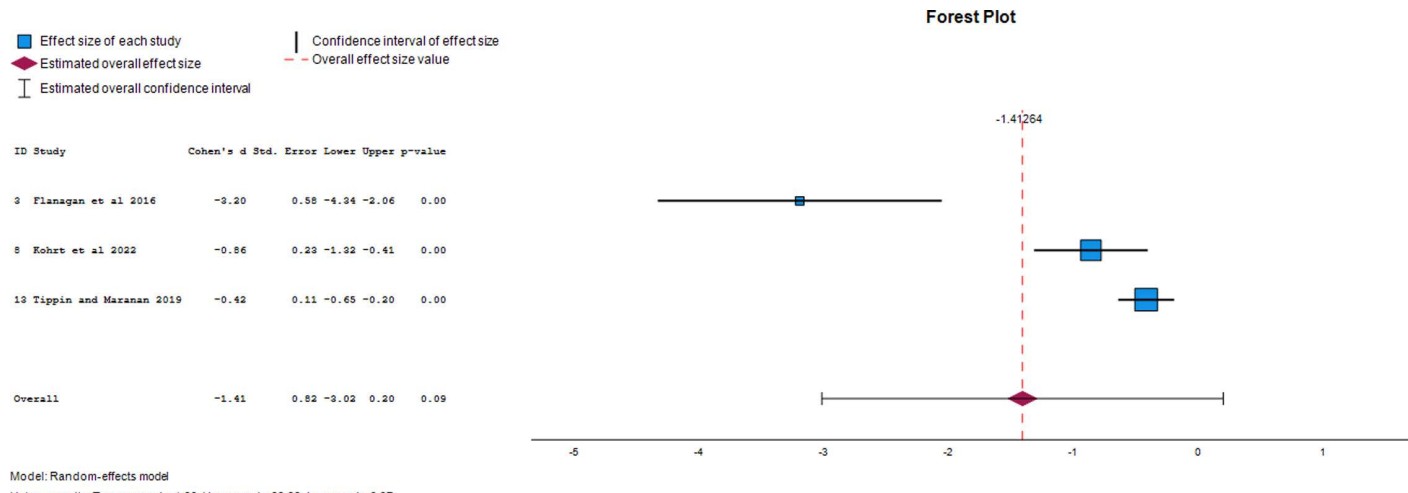

**Fig 8. Forest plot of effect size of photovoice on social distance.**

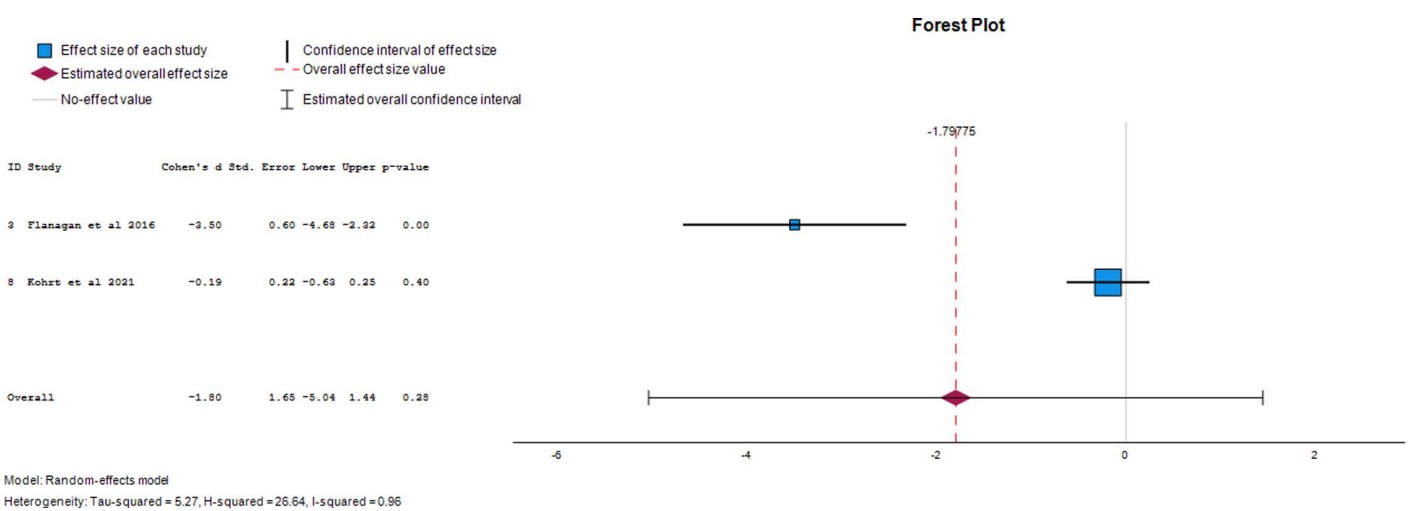

**Fig 9. Forest plot of effect size of photovoice on perception of danger.**

strength-based methods evidenced in the empowering effects of photo-elicitation autonomy in participants, and mental health narrative reconstruction through sharing lived experiences [3,75].

Furthermore, focused group discussions, an integral component of photovoice, foster communication and peer support, which have been shown to reduce social isolation and boost self-esteem [76,77]. These strength-based approaches focus on increasing the resilience in PLWMI through the key characteristics of personal and functional recovery including acceptance, hope, personal distinctiveness, agency/autonomy, social cohesion, engagement, and empowerment [78].

Similarly, the efficacy of adopting participatory research methods such as photovoice by PLWMI within recovery-oriented mental health has been advocated [79–81]. This is facilitated by the sense of functionality photovoice provides, which promotes self-esteem and efficacy, as well as social dimensions of recovery by reducing isolation,

**Table 3. Effect Size of photovoice on mental health and stigma-related outcomes in both PLWMI and HCPs.**

| Outcome | Study ID | Cohen's d | Std. Error | 95% C I | | Heterogeneity |
|---|---|---|---|---|---|---|
| | | | | Lower | Upper | |
| **Mental Health related outcomes in PLWMI** | | | | | | |
| **Depression** | Russionova et al 2014 | -0.33 | 0.16 | -0.63 | 0.03 | Tau-squared = 0.00 H-squared = 1.00 I-squared = 0.00 |
| | Russinova et al 2023 | -0.13 | 0.16 | -0.45 | 0.18 | |
| | Overall | -0.23 | 0.11 | -0.45 | -0.02 | |
| **Recovery** | Zhang et al 2023 | 0.88 | 0.52 | -0.15 | 1.90 | Tau-squared = 0.00 H-squared = 1.00 I-squared = 0.00 |
| | Russinova et al 2014 | 0.16 | 0.15 | -0.14 | 0.46 | |
| | Russinova et al 2023 | 0.17 | 0.16 | -0.15 | 0.48 | |
| | Overall | 0.19 | 0.11 | -0.02 | 0.41 | |
| **Self-Efficacy** | Russinova et al 2014 | 0.27 | 0.15 | -0.04 | 0.57 | Tau-squared = 0.00 H-squared = 1.00 I-squared = 0.00 |
| | Russinova et al 2023 | 0.18 | 0.16 | -0.14 | 0.49 | |
| | Overall | 0.22 | 0.11 | 0.00 | 0.44 | |
| **Social Withdrawal** | Russinova et al 2014 | -0.31 | 0.29 | -0.88 | 0.27 | Tau-squared = 0.00 H-squared = 1.00 I-squared = 0.00 |
| | Russinova et al 2018 | -0.17 | 0.15 | -0.47 | 0.13 | |
| | Russinova et al 2023 | -0.21 | 0.16 | -0.52 | 0.11 | |
| | Overall | -0.20 | 0.10 | -0.41 | 0.00 | |
| **Mental Health related Outcomes in HCPs** | | | | | | |
| **Anger** | Flanagan et al 2016 | -2.00 | 0.46 | -2.91 | -1.09 | Tau-squared = 1.48 H-squared = 14.00 I-squared = 0.93 |
| | Tippin and Maranzan 2019 | -0.22 | 0.11 | -0.44 | 0.01 | |
| | Overall | -1.05 | 0.89 | -2.80 | 0.69 | |
| **Stigma-related outcomes in PLWMI** | | | | | | |
| **Stigma Resistance** | Russionova et al 2014 | -0.16 | 0.29 | -0.73 | 0.41 | Tau-squared = 0.01 H-squared = 1.20 I-squared = 0.17 |
| | Russionova et al 2018 | 0.02 | 0.15 | -0.28 | 0.32 | |
| | Russinova et al 2023 | -0.28 | 0.16 | -0.60 | 0.02 | |
| | Overall | -0.13 | 0.12 | -0.36 | 0.10 | |
| **Stigma-related outcomes in HCPs** | | | | | | |
| **Social Distance** | Flanagan et al 2016 | -3.20 | 0.58 | -4.34 | -2.06 | Tau-squared = 1.90 H-squared = 33.86 I-squared = 0.97 |
| | Tippin and Maranzan 2019 | -0.86 | 0.22 | -1.32 | -0.41 | |
| | Kohrt et al 2021 | -0.42 | 0.11 | -0.65 | -0.20 | |
| | Overall | -1.41 | 0.82 | -3.02 | 0.20 | |
| **Perception of danger** | Kohrt et al 2021 | -3.5 | 0.60 | -4.68 | -2.32 | Tau-squared = 5.27 H-squared = 26.64 I-squared = 0.96 |
| | Flanagan et al 2016 | -0.19 | 0.22 | -0.63 | 0.25 | |
| | Overall | -1.80 | 1.65 | -5.04 | 1.44 | |

fostering community engagement, and increasing the individual's sense of self-worth [79]. This can be seen in a quasi-experimental study conducted with participants from a psychosocial rehabilitation centre which highlights that photovoice facilitates recovery-related outcomes such as empowerment, positive sense of identity, and community integration, and therefore suggests its incorporation into recovery-oriented services of PLWMI [82]. Furthermore, such participatory research enables in-depth communication with the healthcare providers, wherein strength-based experiences beyond living with mental illness can be expressed, such as their interests and contributions to society, which has been shown to significantly reduce healthcare provider stigma, and inadvertently increase the quality of care, and chance of recovery [82].

In contrast, stigma resistance, the ability to challenge mental health stigma, stereotypes, and public prejudice at the personal, peer, and societal level [83], was decreased in this study following the administration of photovoice intervention.

This was measured using the Internalized Stigma of Mental Illness Scale (ISMI [83]), which assesses internalised stigma using five subscales - alienation, stereotype endorsement, discriminatory experiences, social withdrawal, and stigma resistance. The decrease in stigma resistance could be due to psychometric weakness and the low reliability of this subscale in assessing internalized stigma, as reported by its authors [83]. Also, this could be further explained by the short duration of studies analysed for this category, and the increase in participants' sensitivity to stigmatising attitudes in the short term during photovoice intervention [84–86]. However, this meta-analysis revealed that social withdrawal, another subscale of internalised stigma, was reduced in PLWMI following the photovoice intervention. This subscale was shown to have higher internal consistency and test-retest reliability levels when assessing internalised stigma [83]. This is consistent with a previous qualitative study conducted among PLWMI in a mental health club in Hawaii, which identified photovoice as a helpful tool in fostering self-appraisal and inner empowerment, leading to improved social engagement and support [87]. Furthermore, existing research revealed that photovoice fosters empowerment and self-identity, which poses a protective barrier against internalising stigma and discrimination [18]. Likewise, its ability to build resilience, an important factor in mitigating internalised and self-stigma in vulnerable populations has been evidenced [88–91]. Consequently, increased resilience leads to an overall sense of empowerment for the PLWMI, thereby increasing the engagement of PLWMI within communities [77].

Also, self-efficacy, which can be described as the ability of PLWMI to cope and thrive with their mental illness [92], was shown to be enhanced through the influence of photo-elicitation programs in this meta-analysis. This results in a sustained sense of self-worth within the PLWMI, and increased engagement with their community [93,94]. This is evident in a study based on the principles of community-based participatory action research in the United Kingdom, where the participants expounded on the inherent challenges in executing activities of daily living, especially regarding healthy lifestyle choices, due to a perceived inadequacy of measures catering for their needs [95]. The participants elaborated that photovoice availed an opportunity to be consulted for what they needed to lead healthier lives and facilitated the co-creation of strategies with their mental health nurses which enabled the successful execution of healthier behaviours [95]. This offers insight into the challenges of living with mental illness and empowers PLWMI to be advocates for addressing their unmet needs, which promotes a sense of purpose and achievement [92].

Furthermore, exposure to photovoice intervention significantly reduced depression among PLWMI in this meta-analysis. This relates to the findings of a qualitative study among rural adolescents (15–17 years old) with depression and suicidal tendencies in northeast America who experienced a higher sense of self-worth after photovoice intervention [96]. Likewise, the current meta-analysis aligns with the findings of a study in Nepal among twenty-seven 49-year-old rural women with self-reported depression, as measured by the Beck Depression Inventory (BDI), which was found to be significantly reduced after exposure to photovoice intervention [97].

The results of this meta-analysis also suggest that photovoice is effective for reducing social distancing, perception of danger, and anger from healthcare providers towards PLWMI. The visual narratives created can serve as a powerful medium for reducing stereotypes and fostering empathy among the HCPs [98]. Importantly, social distancing from healthcare providers towards PLWMI can intensify the stigma experienced by PLWMI [99]. Two perspectives worth considering for explaining the mechanisms leading to an improvement in these outcomes are the contact-based and humanistic approaches.

One of Allport's seminal criteria for the nature of contact in PLWMI is the perception of equal status between the provider and the receiver [77,100]. Traditional contact-based approaches however disregard the inherent inequality in social status between HCPs and PLWMI [57], explaining why this form of contact produces only short-term effects [101,102]. Photovoice, however, focuses on humanistic approaches, emphasizing empathy, dignity, and respect towards the challenges experienced by PLWMI [103]. In this context, health professionals can relate to the experiences of PLWMI, explaining its remarkable effect on the aforementioned outcomes [103]. Consequently, the balance of power shifts from a feeling of "power over" inherent in healthcare professionals to "power within" in PLWMI [104].

## Implications for policy and practice

This meta-analysis has explored the effectiveness of photovoice on mental health and stigma-related outcomes in PLWMI and their healthcare providers. Building from the findings, photovoice can be implemented in various healthcare settings through targeted interventions that integrate patient experiences into mental health practices. For example, in group therapy or support settings, photovoice can allow participants to document and express personal recovery journeys, transforming personal experiences into visual narratives. This approach enhances therapeutic engagement by providing patients with tangible means to communicate feelings, enhancing self-reflection and enabling the contextualisation of their recovery process [32]. By sharing and discussing their images in group settings, individuals can build a collective sense of understanding and reduce isolation, with interconnected experiences reinforcing shared resilience [105].

In community mental health initiatives, photovoice serves as a potent tool for advocacy and stigma reduction, helping PLWMI address structural challenges and reclaim agency [35]. For instance, public exhibitions of these visual narratives to key stakeholders can highlight issues such as access to resources, social exclusion, or public stigmatisation, challenging stereotypes and promoting empathy among the general population [35]. Consequently, this fosters public dialogue around mental health and reinforces the inclusivity of the PLWMI.

Beyond personal storytelling, this method could be employed in stigma reduction programs within hospitals to strengthen patient-provider relationships [3], as providers gain nuanced insights into patients' lived realities, enhancing empathetic care and creating a more holistic, patient-centred model of mental health support. In outpatient settings, photovoice can serve as a means of monitoring progress by capturing visual narratives that inform clinical discussions [3]. Through the implementation of this intervention in such settings, HCPs can create a more holistic and patient-centred approach to mental health care, leading to improved personal recovery outcomes and the elimination of stigma.

## Limitations of the study

Heterogeneity was greater than 50% in three outcomes, anger (0.93), perception of danger (0.96), and social distance (0.99) which demonstrated high heterogeneity. The heterogeneity was attributed to the small number of included studies in the meta-analysis for these outcomes [106]. This is because while the "$I^2$" provides an estimate, it does not eradicate the uncertainty that arises from having a limited number of studies in a meta-analysis [107]. Therefore, it is important to interpret the heterogeneity encountered with caution. In addition, the variability in the sample sizes of the studies utilised in the meta-analysis is another potential contributing factor to the high heterogeneity. For example, for anger, there were 27 participants in the study conducted by Flanagan *et al.* compared to 303 in the study conducted by Tippin and Maranzan. Furthermore, differences in the design and administration techniques of the photovoice intervention are other potential sources of the heterogeneity encountered within this meta-analysis. However, as advised by Cochrane, a sub-group analysis or meta-regression test was not conducted due to less than 10 studies being analysed for each outcome [108]. Future RCTs would provide opportunities for the inclusion of a higher number of studies in subsequent meta-analyses to generate more precise effect estimates.

Nonetheless, most of the included studies were of moderate quality, with one study of high quality, and two studies having low quality. Future RCTs should therefore consider adopting rigorous and unbiased study designs to ensure studies available for review are of the highest quality. The language restriction to studies exclusively published in the English Language could have inadvertently led to language bias, in which useful evidence may have been excluded, potentially impacting the generalisability of the study's findings. However, this was done in this meta-analysis to ensure efficient management of resources. Future systematic reviews should therefore expand its inclusion criteria to include all languages. Additionally, future RCTs should consider exhausting all outcomes of mental illness and stigma, extending to outcomes not included in this meta-analysis. As mental healthcare continues to evolve, further research and the successful implementation of photovoice strategies are essential for promoting positive mental health outcomes for diverse populations.

## Conclusion

Overall, it can be deduced that photovoice does have a significant effect on improving the mental health outcomes of PLWMI. Furthermore, photovoice significantly reduces the negative mental health and stigma-related outcomes in HCPs that have consequences for the mental health of PLWMI. Therefore, it can be concluded that photovoice improves the mental health outcomes of PLWMI and has a positive effect on the mental health and stigma-related behaviours of HCPs towards PLWMI. Conversely, the findings suggest that photovoice reduces the positive stigma-related outcomes in PLWMI, such as stigma resistance in the short term. Understanding the effect of photovoice on stigma resistance warrants further research that is conducted over longer study durations.

## Supporting information

**S1 Table. Full search history on all databases.**
(DOCX)

**S2 Table. Data extraction form.**
(DOCX)

**S3 Table. Full data extraction table for all included studies (PLWMI).**
(XLSX)

**S4 Table. Full data extraction table for all included studies (HCPs).**
(XLSX)

**S1 Appendix. PRISMA 2020 checklist.**
(DOCX)

**S2 Appendix. Reference lists of included studies.**
(DOCX)

**S3 Appendix. Funnel plots for all outcomes.**
(DOCX)

**S4 Appendix. Studies identified in the literature search; Excluded studies with reasons.**
(XLSX)

## Acknowledgments

We acknowledge the willingness and commitment by team members to achieve this milestone.

## Author contributions

**Conceptualization:** Adeniyi Adeboye, Judy Wasige.

**Data curation:** Adeniyi Adeboye, Whitney Onuorah, Chizor Samuel-Nwokeji, Chimaobi Nwanguma, Adekunle Akerele.

**Formal analysis:** Adeniyi Adeboye, Ugochukwu Aghalu, Chimaobi Nwanguma, Adekunle Akerele.

**Funding acquisition:** Judy Wasige.

**Investigation:** Adeniyi Adeboye, Whitney Onuorah, Chizor Samuel-Nwokeji.

**Methodology:** Adeniyi Adeboye, Whitney Onuorah, Chizor Samuel-Nwokeji, Chimaobi Nwanguma.

**Project administration:** Adeniyi Adeboye, Judy Wasige.

**Resources:** Judy Wasige.

**Software:** Adeniyi Adeboye, Ugochukwu Aghalu, Adekunle Akerele.

**Supervision:** Adeniyi Adeboye, Judy Wasige.

**Validation:** Adeniyi Adeboye, Whitney Onuorah, Chimaobi Nwanguma, Adekunle Akerele.

**Visualization:** Adeniyi Adeboye, Ugochukwu Aghalu, Whitney Onuorah, Chizor Samuel-Nwokeji, Adekunle Akerele, Judy Wasige.

**Writing – original draft:** Adeniyi Adeboye, Ugochukwu Aghalu, Whitney Onuorah, Chizor Samuel-Nwokeji.

**Writing – review & editing:** Ugochukwu Aghalu, Whitney Onuorah, Chizor Samuel-Nwokeji, Adekunle Akerele, Judy Wasige.

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
