## [Decision Letter · Decision Letter 0]

PGPH-D-24-00689

The effectiveness of Photovoice in mental health: A systematic review and meta-analysis.

Dear Dr. Aghalu,

Thank you for submitting your manuscript to PLOS Global Public Health. After careful consideration, we feel that it has merit but does not fully meet PLOS Global Public Health’s publication criteria as it currently stands. Therefore, we invite you to submit a revised version of the manuscript that addresses the points raised during the review process.

Your manuscript has been evaluated by two reviewers, and their comments are appended below.

Both reviewers have identified concerns with your manuscript, particularly regarding the study design, reporting, and results. Please ensure you address each of the reviewers' comments when revising your manuscript.

Please note that one or more reviewers has recommended that you cite specific previously published works. As always, we recommend that you please review and evaluate the requested works to determine whether they are relevant and should be cited. It is not a requirement to cite these works. We appreciate your attention to this request.

We look forward to receiving your revised manuscript.

Kind regards,

Hugh Cowley

Staff Editor

Journal Requirements:

1. We ask that a manuscript source file is provided at Revision. Please upload your manuscript file as a .doc, .docx, .rtf or .tex.

Additional Editor Comments (if provided):

Reviewers' comments:

Reviewer's Responses to Questions

**Comments to the Author**

1. Does this manuscript meet PLOS Global Public Health’s publication criteria ? Is the manuscript technically sound, and do the data support the conclusions? The manuscript must describe methodologically and ethically rigorous research with conclusions that are appropriately drawn based on the data presented.

Reviewer #1: Partly

Reviewer #2: Partly

2. Has the statistical analysis been performed appropriately and rigorously?

Reviewer #1: I don't know

Reviewer #2: I don't know

3. Have the authors made all data underlying the findings in their manuscript fully available (please refer to the Data Availability Statement at the start of the manuscript PDF file)?

Reviewer #1: Yes

Reviewer #2: Yes

4. Is the manuscript presented in an intelligible fashion and written in standard English?

Reviewer #1: Yes

Reviewer #2: Yes

5. Review Comments to the Author

Reviewer #1: The authors have attempted to highlight a very interesting and novel intervention “photovoice” and its effects especially in PLWMI’s recovery and stigma experiences. While the study topic is interesting, there are some key issues that needs to be addressed by authors for the conceptual clarity and to make the findings relevant.

Abstract: Please follow Prisma 2020 guidelines for the reporting of systematic review abstract.

Pg4, p2: study evaluates the impact of photovoice in mental ill health: vague and not clear based on included papers. The abstract mentions the evaluation of the effects of photovoice on 8 mental health outcomes. For clarity, it would be helpful if authors separate mental health outcomes and stigma outcomes as different.

Pg5 p2: “what is the impact of photovoice on mental health outcomes?” – Similar to above comment, it would be great if authors could modify this into the impact of photovoice on mental health and stigma outcomes of PLWMI and HCPs.

Pg6: Search strategy- The authors need to include full search strategies used for all databases including any filters and limits used. (See Prisma guideline for reporting)- I saw that authors have included prisma 2009 checklist- this need to be referenced in the main manuscript and as there is prisma guideline/checklist for 2020, it would be great if authors updated the checklist.

Table 1: Characteristics of included studies- it is confusing as some studies only had PLWMIs some only had HCPs while some had both as population and it is also not clear which outcomes/questionnaires were used for which population? Would be helpful if the authors added a column on population for the studies.

Another useful thing to mention in the table is the outcome measure as per your analysis. Currently the outcomes listed in the table does not match with the outcomes you report. How were the listed outcomes such as dangerousness, social desirability, empowerment, coping with stigma etc categorized into the outcomes you report on later?

Pg 19, 20: The data analysis section should go into the methods section rather than the results.

Pg 21: The effects of photovoice on stigma resistance and social distance of PLWMIs is not clear. Social distance is used to measure public/provider stigma towards PLWMIs – not with PLWMIs. Is this social withdrawal instead of social distance?

Also, how was stigma resistance measured- I don’t see stigma resistance mentioned in any of the outcomes or questionnaires you have listed in Table 1? Is this stigma coping?

The papers listed under PLWMI outcomes of anger, recovery, social distance, etc. (Flanagan 2016, Tippin and Maranzan 2019) are targeting health workers rather than PLWMI- I am not sure how the authors assigned the health workers' outcomes to PLWMIs?

On a similar note, the paper Zhang et al 2023, the study uses mindfulness-based psychoeducation as an intervention with a component of photovoice and its effects on YAIR. The author’s interpretation that this is the effect of photovoice rather than the effect of mindfulness-based psychoeducation might be a bit farfetched.

Pg 25 I think there were a lot more papers on health worker stigma outcomes- as mentioned in the study characteristics (table 1), however, the authors have only described the effects of Tergesen's 2021 paper on the effect of photovoice on the healthcare provider section. In addition, the Tergesen paper mentions an increase in stigma scores among medical students after watching the psychosis photovoice video. This is not reflected in the findings of the current paper.

Reviewer #2: Thanks for the opportunity to review this paper. The paper aims to understand the effect of PhotoVoice among people with lived experiences of mental health and health care providers. Though the topic of this paper is very important, especially considering increased call to engage with people with lived experience in mental health research and interventions and PhotoVoice being one of the ways to do that, the paper requires revision. More specific comments are below.

A. Introduction: The authors talk about the rationale and advantages of Photovoice but more detail on the content and process of Photovoice is required – e.g. how are the participants selected, trained, how do they write stories and take their photos and how are they presented across different settings. This should help in setting up the background of the study.

“Mental health outcomes” – that the author describes in the abstract and rest of the paper should be described in detail. E.g. having depression (a mental health condition) and self-efficacy under the same domain is confusing. A clear operational definition is required.

B. Results: Need more conceptual clarity on the eight outcomes. e.g. Recovery is a very vague term that means different thing to different people. The authors note that in the discussion, but they are also requested to discuss how it was conceptualized in the studies they reviewed. This should help in making the results more cohesive.

Social distance scale is also often used as a measure of stigma – can the authors discuss how the studies have used/defined it so that it can be presented as a different domain from stigma?

Table 1 needs more clarity – E.g. Zhang 2023: uses social distance scale but the outcome is better caregiving experience and better recovery. In Russinova 2014 the questionnaire used by the study is ISMI and coping with stigma, but outcome is more than that including depression. In the table, it is important to note what measure was used for depression? This makes reading the table confusing. The authors are requested to revise the table making it more specific and detail and subsequently reflect that while describing the result section.

C. Discussion: The authors have initiated good discussion points but needs to be tighter. Some suggested papers to support this section and introduction.

a. On methods:

• Abayneh 2022 paper: https://link.springer.com/article/10.1186/s12913-022-08290-x

• Rai 2023 paper: https://www.cambridge.org/core/journals/global-mental-health/article/photovoice-method-for-collaborating-with-people-with-lived-experience-of-mental-health-conditions-to-strengthen-mental-health-services/4FA9A30F1C6DF9F07FE9F952403F20FB

b. On theory: Kohrt 2020 paper https://www.sciencedirect.com/science/article/abs/pii/S027795362030071X

D. Other comments

a. One study to my knowledge I see missing here, though cited is Kohrt 2021 JAMA paper on RESHAPE RCT – doi: 10.1001/jamanetworkopen.2021.31475

Was it excluded because it was a pilot RCT and not a full RCT?

6. PLOS authors have the option to publish the peer review history of their article (what does this mean? ). If published, this will include your full peer review and any attached files.

**Do you want your identity to be public for this peer review?** For information about this choice, including consent withdrawal, please see our Privacy Policy .

Reviewer #1: No

Reviewer #2: No

---

## [Decision Letter · Decision Letter 1]

PGPH-D-24-00689R1

The impact of photovoice on mental health and stigma: A systematic review and meta-analysis.

Dear Dr. Aghalu,

Thank you for submitting your manuscript to PLOS Global Public Health. After careful consideration, we feel that it has merit but does not fully meet PLOS Global Public Health’s publication criteria as it currently stands. Therefore, we invite you to submit a revised version of the manuscript that addresses the points raised during the review process.

We look forward to receiving your revised manuscript.

Kind regards,

Feten Fekih-Romdhane

Academic Editor

Journal Requirements:

Additional Editor Comments (if provided):

Reviewers' comments:

Reviewer's Responses to Questions

**Comments to the Author**

1. If the authors have adequately addressed your comments raised in a previous round of review and you feel that this manuscript is now acceptable for publication, you may indicate that here to bypass the “Comments to the Author” section, enter your conflict of interest statement in the “Confidential to Editor” section, and submit your "Accept" recommendation.

Reviewer #3: All comments have been addressed

Reviewer #4: All comments have been addressed

2. Does this manuscript meet PLOS Global Public Health’s publication criteria ? Is the manuscript technically sound, and do the data support the conclusions? The manuscript must describe methodologically and ethically rigorous research with conclusions that are appropriately drawn based on the data presented.

Reviewer #3: Yes

Reviewer #4: Yes

3. Has the statistical analysis been performed appropriately and rigorously?

Reviewer #3: Yes

Reviewer #4: Yes

4. Have the authors made all data underlying the findings in their manuscript fully available (please refer to the Data Availability Statement at the start of the manuscript PDF file)?

Reviewer #3: Yes

Reviewer #4: Yes

5. Is the manuscript presented in an intelligible fashion and written in standard English?

Reviewer #3: Yes

Reviewer #4: Yes

6. Review Comments to the Author

Reviewer #3: I congratulate the authors for addressing adequately the comments raised in the previous review round.

Upon reviewing the references, I have observed ref number 72 to be incorrectly cited The words "Link to external site this link will open in a new window" should be omitted.

Reviewer #4: >Major Points:

Relevance and Novelty:

Your study is highly relevant and novel, particularly in its exclusive focus on photovoice's impact on both PLWMI and HCPs. It fills a critical gap in the existing literature. However, I suggest emphasizing this point more clearly in the introduction, especially when discussing the study's significance and how it advances the current knowledge.

Clarification of Terms and Concepts:

Some key concepts, such as personal recovery versus functional recovery, would benefit from clearer definitions and distinctions. This will help readers unfamiliar with these terms better understand the implications of your findings.

Heterogeneity and Study Quality:

While you acknowledge the high heterogeneity in certain outcomes (e.g., anger, perception of danger, and social distance), a more detailed discussion on how this heterogeneity might affect the interpretation of your findings would be beneficial. Additionally, consider discussing potential strategies to address this issue in future research.

Discussion of Stigma Resistance:

The reduction in stigma resistance following photovoice intervention is an intriguing and somewhat counterintuitive finding. The explanation provided—related to the short duration of the studies analyzed—could be expanded. I recommend exploring alternative explanations or discussing the potential impact of the intervention's context on this outcome.

Implications for Practice:

While you provide a solid discussion on the implications for mental health services, more specific examples or recommendations on how photovoice can be integrated into current practices would enhance the practical utility of your findings. This could include concrete steps for implementation in various healthcare settings.

>Minor Points:

Formatting:

Ensure consistency in the formatting of tables and figures. Some figures appear to have formatting issues that should be corrected to improve the overall presentation.

Literature Review:

The introduction could be strengthened by providing a brief overview of key studies that have previously explored photovoice’s impact on mental health. This would establish a stronger foundation for your study and highlight the specific gaps your research addresses.

Statistical Analysis:

The choice of random-effects models and the handling of potential publication bias are appropriate, but further clarification in these areas would add to the robustness of your analysis. Consider providing a brief explanation of these choices in the methods or results section.

I recommend minor revisions to address these points for the final version of your manuscript.

7. PLOS authors have the option to publish the peer review history of their article (what does this mean? ). If published, this will include your full peer review and any attached files.

**Do you want your identity to be public for this peer review?** For information about this choice, including consent withdrawal, please see our Privacy Policy .

Reviewer #3: No

Reviewer #4: **Yes: ** Khushbu Balsara

---

## [Decision Letter · Decision Letter 2]

PGPH-D-24-00689R2

The impact of photovoice on mental health and stigma: A systematic review and meta-analysis.

Dear Dr. Aghalu,

Thank you for submitting your manuscript to PLOS Global Public Health. After careful consideration, we feel that it has merit but does not fully meet PLOS Global Public Health’s publication criteria as it currently stands. Therefore, we invite you to submit a revised version of the manuscript that addresses the points raised during the review process.

We look forward to receiving your revised manuscript.

Kind regards,

Feten Fekih-Romdhane

Academic Editor

Journal Requirements:

Additional Editor Comments (if provided):

Reviewers' comments:

Reviewer's Responses to Questions

**Comments to the Author**

1. If the authors have adequately addressed your comments raised in a previous round of review and you feel that this manuscript is now acceptable for publication, you may indicate that here to bypass the “Comments to the Author” section, enter your conflict of interest statement in the “Confidential to Editor” section, and submit your "Accept" recommendation.

Reviewer #4: All comments have been addressed

Reviewer #5: (No Response)

Reviewer #6: All comments have been addressed

2. Does this manuscript meet PLOS Global Public Health’s publication criteria ? Is the manuscript technically sound, and do the data support the conclusions? The manuscript must describe methodologically and ethically rigorous research with conclusions that are appropriately drawn based on the data presented.

Reviewer #4: Yes

Reviewer #5: Yes

Reviewer #6: Partly

3. Has the statistical analysis been performed appropriately and rigorously?

Reviewer #4: N/A

Reviewer #5: Yes

Reviewer #6: I don't know

4. Have the authors made all data underlying the findings in their manuscript fully available (please refer to the Data Availability Statement at the start of the manuscript PDF file)?

Reviewer #4: Yes

Reviewer #5: Yes

Reviewer #6: Yes

5. Is the manuscript presented in an intelligible fashion and written in standard English?

Reviewer #4: Yes

Reviewer #5: Yes

Reviewer #6: No

6. Review Comments to the Author

Reviewer #4: (No Response)

Reviewer #5: Formatting Issues

On page 17, fig. 2 can fit in the space and it does not need to be on a separate page.

On page 20, the paragraph before "Depression" has an unwanted spacing

On page 21, in the section on "Stigma Resistance", there should be spacing after "-0.13". Same on page 22 for "0.27"

On the Reference page,

Ref 61 needs fixing.

Ref 37, 49, and 55, links to external sites opened a new window that is not related to the cited paper.

On Implications for policy and practice and limitations to the study

Specific examples such as how health professionals can utilize photovoice in therapy sessions or support groups can be pointed out. Another example can be how to implement photovoice in community mental health programs.

Providing a comprehensive discussion of unexpected findings that is not only limited to the duration of studies but also other specific reasons why there was a reduced stigma resistance will make the result more robust and insightful.

I am aware this review is for the final manuscript, but these minor revisions need to be addressed and I would like to commend the authors for the great effort that was put into addressing previous comments raised.

Reviewer #6: Dear Editor Dr. Feten Fekih-Romdhane

Dear Authors

I hope this message finds you well. Apart from thanking you for allowing me to review the manuscript "The impact of photovoice on mental health and stigma: A systematic review and meta-analysis", I would like to point out some points that could be contributed.

The introduction could have an order that better contextualizes the study. There is a lot of data on the advantages of this innovation, it is undoubtedly a great tool for recovery, for validating rights, for building mental health in the first person. However, in this section the reading is not very clear, especially for someone who is not an expert in photovoice.

As a suggestion, the introduction could address: What is photovoice? What are its components?, How could it generate so many benefits? In addition to the benefits in mental health in general and in specific pathologies. It is also not clear how the authors make a transition to HIV, when it is not an evident category either in the abstract or in the previous paragraphs.

At a methodological level, I find it a solid study. However, I have limitations when viewing the figures. Figure one is vague, but that will be organized later, I imagine. However, the other figures cannot be seen in the version that I am reviewing, and this prevents the rigor that is described from being specified.

Sometimes the difference with a scoping review is not clear. Making this difference explicit could strengthen the methodology: Munn, Z., Peters, M.D.J., Stern, C. et al. Systematic review or scoping review? Guidance for authors when choosing between a systematic or scoping review approach. BMC Med Res Methodol 18, 143 (2018). https://doi.org/10.1186/s12874-018-0611-x.

Please indicate the languages of the study. Also, why other databases were not included, especially Latin American studies (Redalyc, Scielo).

The categories of organization in the results are confusing. Depression and self-efficacy are combined in the same category. Organizing them better would give greater force to these very interesting results.

The discussion could be articulated with a better organized introduction, to show novel results that move the barriers of knowledge. For example, what are the mechanisms of the results indicated in the introduction, how does it improve stigma… Given that it is a participatory process by definition, how do you define effectiveness?

I find this to be a valuable and innovative study. I suggest some adjustments to make it readable to a global audience that could possibly benefit from this intervention/action.

7. PLOS authors have the option to publish the peer review history of their article (what does this mean? ). If published, this will include your full peer review and any attached files.

**Do you want your identity to be public for this peer review?** For information about this choice, including consent withdrawal, please see our Privacy Policy .

Reviewer #4: No

Reviewer #5: No

Reviewer #6: **Yes: ** Felipe Agudelo-Hernández

---

## [Decision Letter · Decision Letter 3]

PGPH-D-24-00689R3The impact of photovoice on mental health and stigma: A systematic review and meta-analysis.PLOS Global Public Health

Dear Dr. Aghalu,  Thank you for submitting your manuscript to  PLOS Global Public Health., and for responding to our recent requests regarding your submission. Unfortunately, in our final editorial checks of the documents that you supplied, we have concluded that your submission does not comply with our policies around data availability. We are therefore overturning the provisional editorial accept decision, and rejecting this manuscript.  PLOS journals require authors to make all data necessary to replicate their study’s findings publicly available without restriction at the time of publication (https://journals.plos.org/plosone/s/data-availability). In this case, the following underlying data were not provided as requested: A numbered table of all studies identified in the literature search, including those that were excluded from the analyses.   As a result of these concerns, we cannot consider the manuscript for publication. I am very sorry that this issue was identified at such a late stage.   

Yours sincerely,

Jennifer Tucker, PhD

Staff Editor

Additional Editor Comments (if provided):

Reviewers' comments:

Reviewer's Responses to Questions**Comments to the Author**

1. If the authors have adequately addressed your comments raised in a previous round of review and you feel that this manuscript is now acceptable for publication, you may indicate that here to bypass the “Comments to the Author” section, enter your conflict of interest statement in the “Confidential to Editor” section, and submit your "Accept" recommendation.Reviewer #5: All comments have been addressedReviewer #6: All comments have been addressed

2. Does this manuscript meet PLOS Global Public Health’s publication criteria ? Is the manuscript technically sound, and do the data support the conclusions? The manuscript must describe methodologically and ethically rigorous research with conclusions that are appropriately drawn based on the data presented.Reviewer #5: YesReviewer #6: Partly

3. Has the statistical analysis been performed appropriately and rigorously?Reviewer #5: N/AReviewer #6: N/A

4. Have the authors made all data underlying the findings in their manuscript fully available (please refer to the Data Availability Statement at the start of the manuscript PDF file)?

The PLOS Data policy requires authors to make all data underlying the findings described in their manuscript fully available without restriction, with rare exception. The data should be provided as part of the manuscript or its supporting information, or deposited to a public repository. For example, in addition to summary statistics, the data points behind means, medians and variance measures should be available. If there are restrictions on publicly sharing data—e.g. participant privacy or use of data from a third party—those must be specified.Reviewer #5: YesReviewer #6: Yes

5. Is the manuscript presented in an intelligible fashion and written in standard English?

PLOS Global Public Health does not copyedit accepted manuscripts, so the language in submitted articles must be clear, correct, and unambiguous. Any typographical or grammatical errors should be corrected at revision, so please note any specific errors here.Reviewer #5: YesReviewer #6: Yes

6. Review Comments to the Author

Please use the space provided to explain your answers to the questions above. You may also include additional comments for the author, including concerns about dual publication, research ethics, or publication ethics. (Please upload your review as an attachment if it exceeds 20,000 characters)Reviewer #5: Authors have addressed all concerns and corrected all errors as instructed.Reviewer #6: I have no further comments for this manuscript

7. PLOS authors have the option to publish the peer review history of their article (what does this mean? ). If published, this will include your full peer review and any attached files.

**Do you want your identity to be public for this peer review?** For information about this choice, including consent withdrawal, please see our Privacy Policy .Reviewer #5: NoReviewer #6: No

---

## [Decision Letter · Decision Letter 4]

The impact of photovoice on mental health and stigma: A systematic review and meta-analysis.

PGPH-D-24-00689R4

Dear Dr. Aghalu,

We are pleased to inform you that your manuscript 'The impact of photovoice on mental health and stigma: A systematic review and meta-analysis.' has been provisionally accepted for publication in PLOS Global Public Health.

Best regards,

Julia Robinson

Executive Editor

Reviewer Comments (if any, and for reference):

Reviewer's Responses to Questions

**Comments to the Author**

1. If the authors have adequately addressed your comments raised in a previous round of review and you feel that this manuscript is now acceptable for publication, you may indicate that here to bypass the “Comments to the Author” section, enter your conflict of interest statement in the “Confidential to Editor” section, and submit your "Accept" recommendation.

Reviewer #5: All comments have been addressed

2. Does this manuscript meet PLOS Global Public Health’s publication criteria ? Is the manuscript technically sound, and do the data support the conclusions? The manuscript must describe methodologically and ethically rigorous research with conclusions that are appropriately drawn based on the data presented.

Reviewer #5: Yes

3. Has the statistical analysis been performed appropriately and rigorously?

Reviewer #5: Yes

4. Have the authors made all data underlying the findings in their manuscript fully available (please refer to the Data Availability Statement at the start of the manuscript PDF file)?

Reviewer #5: Yes

5. Is the manuscript presented in an intelligible fashion and written in standard English?

Reviewer #5: Yes

6. Review Comments to the Author

Reviewer #5: All comments have been thoroughly addressed.

7. PLOS authors have the option to publish the peer review history of their article (what does this mean? ). If published, this will include your full peer review and any attached files.

**Do you want your identity to be public for this peer review?** For information about this choice, including consent withdrawal, please see our Privacy Policy .

Reviewer #5: No
